# The Therapeutic Wound Healing Bioactivities of Various Medicinal Plants

**DOI:** 10.3390/life13020317

**Published:** 2023-01-23

**Authors:** Ghosoon Albahri, Adnan Badran, Akram Hijazi, Anis Daou, Elias Baydoun, Mohamad Nasser, Othmane Merah

**Affiliations:** 1Platform de Recherche et D’analyse en Sciences de L’environnement (EDST-PRASE), Beirut 1107, Lebanon; 2Department of Nutrition, University of Petra, Amman 1196, Jordan; 3Pharmaceutical Sciences Department, College of Pharmacy, QU Health, Qatar University, Doha P.O. Box 2713, Qatar; 4Department of Biology, American University of Beirut, Beirut 1107, Lebanon; 5Laboratoire de Chimie Agroindustrielle (LCA), Université de Toulouse, INRA, INPT, 31030 Toulouse, France; 6Département Génie Biologique, IUT A, Université Paul Sabatier, 32000 Auch, France

**Keywords:** wounds, wound healing, bioactive components, medicinal plants

## Abstract

The skin serves as the body’s first line of defense, guarding against mechanical, chemical, and thermal damage to the interior organs. It includes a highly developed immune response that serves as a barrier against pathogenic infections. Wound healing is a dynamic process underpinned by numerous cellular activities, including homeostasis, inflammation, proliferation, and remodeling, that require proper harmonious integration to effectively repair the damaged tissue. Following cutaneous damage, microorganisms can quickly enter the tissues beneath the skin, which can result in chronic wounds and fatal infections. Natural phytomedicines that possess considerable pharmacological properties have been widely and effectively employed forwound treatment and infection prevention. Since ancient times, phytotherapy has been able to efficiently treat cutaneous wounds, reduce the onset of infections, and minimize the usage of antibiotics that cause critical antibiotic resistance. There are a remarkable number of wound-healing botanicals that have been widely used in the Northern Hemisphere, including *Achiella millefolium*, *Aloe vera*, *Althaea officinalis*, *Calendula officinalis*, *Matricaria chamomilla*, *Curcuma longa*, Eucalyptus, Jojoba, plantain, pine, green tea, pomegranate, and Inula. This review addresses the most often used medicinal plants from the Northern Hemisphere that facilitate the treatment of wounds, and also suggests viable natural alternatives that can be used in the field of wound care.

## 1. Introduction

The entire body’s surface is covered by the intricate organ known as skin. It acts as both a physical shield and a barrier between the body and the outside environment, preventing the loss of water and electrolytes, limiting chemical penetration, and guarding against pathogenic microbes [1]. In developed nations, wounds end up costing millions of dollars each year as they are a major public health concern due to the microbiological complications they cause, such as local or apparent infection, poor healing, and the development of multi-resistant bacteria [2]. Traumatic wounds occur particularly in demographics of people at the two opposite ends of the age spectrum, namely infants and the elderly [3]. As the population ages, chronic lower limb wounds place a greater burden on healthcare services [4]. Acute wounds tend not to disrupt the long-lasting restoration of the anatomical and functional integrity of the skin, whereas chronic wounds can cause the healing process to fail in an arranged manner [5]. The wound healing process involves the interconnected movements of numerous cell types with different functions during the stages of homeostasis, inflammation, proliferation, re-epithelialization, and remodeling, as is shown in Figure 1.

Starting from the homeostasis phase, damaged blood vessels rapidly contract after injury, and a blood clot forms to stop exsanguination caused by vascular damage; platelets are then triggered when they come into contact with the vascular sub-endothelial matrix, homeostasis, and coagulation [7,8]. Platelets are packed with cytokines and growth factors, including (i) insulin-like, (ii) platelet-derived, (iii) transforming, and (iv) epidermal growth factors. These chemicals activate and draw neutrophils, which later attract macrophages, endothelial cells, and fibroblasts, acting as repair agents in the wound healing cascade [9]. Next is the inflammatory phase, which starts with the early inflammatory response that begins during the late phase of coagulation and ends shortly thereafter. This phase triggers neutrophil aggregation in the wound area, where their primary function is to operate as wound cleaners. Various chemo-attractive chemicals are secreted by bacteria and platelet products within 24 to 36 h after injury; these draw neutrophils to the wound area in order to remove the bacteria via phagocytosis and in so doing prevent infection. Once all contaminating microorganisms have been eradicated, neutrophils are eliminated by apoptosis; after 48–72 h, wound macrophages continue the phagocytosis process. Macrophages have a longer lifespan than neutrophils and operate as essential regulatory cells, supplying an ample reservoir of potent tissue growth factors, fibroblasts, and endothelial cells. Thus, the removal of dead tissue and foreign bodies by neutrophils and macrophages results in the production of growth factors, cytokines, proteases, reactive oxygen species, and coagulation factors [10]. The extracellular matrix (ECM) basement membrane is rebuilt by keratinocytes and fibroblasts, which are crucial to the development of the granulation tissue that closes the wound and serves as a structural platform for cell adhesion, migration, growth, and differentiation as the wound moves toward the completion of repair [11]. The primary cell type involved in remodeling the wound’s ECM is fibroblasts that create mature collagen fibrils [12]. There are several impediments to the wound healing process, including foreign bodies bringing the possibility of infection, ischemia, edema or elevated pressure [13]. Thus, to encourage wound healing and prevent additional problems, appropriate diagnosis and treatment are crucial [14].

One of the fundamental steps in wound treatment is preserving a clean wound bed. If done correctly, wound washing can lessen the bioburden and postpone the formation of biofilms. Although antiseptic solutions are used to clean wounds in order to avoid infection, it is not generally known whether antiseptic solutions such as irrigation liquids speed up the healing process [15]. An improper healing procedure—which can result in significant harm, including skin loss and the start of an infection, typically in the case of chronic wounds—is the most frequent and unavoidable barrier to wound healing [16].

Herbal medicines (HM), which are known as complementary and alternative medicines, have been used over the decades to treat medical ailments and promote wellness through their bioactive ingredients [17]. Over time, humans have discovered which plant species are more effective as treatments for specific illnesses. The use of herbal medicine is a standard practice in traditional Chinese Medicine, Ayurveda, Unani, Russian herbalism, and other medical systems to apply botanicals topically to treat wounds and other dermatological problems (Table 1). Moreover, the biological functions of botanicals’ secondary metabolites are what give rise to their pharmacological effects [18].

Many nations and international organizations have included a one-health strategy in their action plans to deal with antibiotic resistance, where improvements in antimicrobial usage policies/regulations, infection control, sanitation and alternatives to antimicrobials are all necessary measures [23]. An increased likelihood of unsuccessful treatment and recurrent infections is linked to persistent antimicrobial resistance. As a result, they play a significant role in rising mortality rates, which raise healthcare expenses. Antibiotic resistance is easily detectable using common microbiological tests, and the harm it poses has long been understood. Antibiotic persistence is a phenomenon where bacteria survive antibiotic exposure despite being completely sensitive. Unlike antibiotic resistance, antibiotic persistence is hard to measure and is sometimes overlooked, which could result in treatment failure [24]. Wounds have become a category in the National Institutes of Health’s Research Portfolio Online Reporting Tool—this is owing to rising health care expenses, an ageing population, increasing awareness of infection hazards (including biofilms) that are challenging to treat, and the ongoing danger of diabetes and obesity worldwide. Moreover, by 2024 the market for wound care products is anticipated to grow economically to around $15–22 billion annually [25]. The implementation of measures designed to lessen the development of antibiotic resistance and the dissemination of resistant bacteria must be enforced. Thus, over the years, plant-based medications have been used to treat a variety of medical disorders [26] and diverse skin diseases such as atopic dermatitis, acne vulgaris, and psoriasis [27].

There are sequential steps for any drug’s discovery, development, and approval [28]. Many laboratories throughout the world are concerned about the screening and testing of extracts against various pharmacological targets in order to profit from the immense natural chemical diversity [29]. The current review aims to introduce the main and most effective botanicals and herbal treatments and their modes of action intreating wounds. The purpose of this research is to provide dermatologists and scientists with the most effective natural substitutes for the wound care sector.

## 2. Plant Species and Their Potential Therapeutic Interests

### 2.1. Achilleamillefolium L.

Numerous yarrow (*Achillea*) species have been used in ethnopharmacology for quite some time. *A. millefolium* is among the most frequently utilized natural plant species for treating wounds, bleeding, stomachaches, gastrointestinal disorders, colds, flu, and stomach issues [30]. The Greek hero Achilles is reported to have utilized this plant to monitor his blood flow and to cure his wounds during the Trojan War. Thus, the genus name may have been derived from his name, while the species name “*millefolium*” alludes to the delicately split, feather-like leaves [30].

Since ancient times, several *Achillea* species have been utilized as traditional herbal medicines in numerous cultures. One of the most significant and commonly used medicinal plants in the world is yarrow, *Achillea millefolium* L., which grows naturally throughout Europe, Asia, North Africa, and North America [31]. The flowery plant is indigenous to the Northern Hemisphere and has been employed in conventional medicine as an astringent, antiseptic, anti-inflammatory, and antispasmodic agent to speed up the recovery of burns, ulcers, cuts, and wounds [32].

The primary *Achillea*’s antioxidant phytochemical components are presented in Figure 2.

The antioxidant and anti-inflammatory benefits of *A. millefolium* have been linked to its flavonoid concentration [33]. The considerable tyrosinase inhibiting, antioxidant, and antibacterial activities of *Achillea* extracts make them intriguing candidates for use as active ingredients in pharmaceuticals and cosmetics that protect the skin from the damaging effects of environmental stresses [34]. In cultured skin biopsies, the expression profiles of cytokeratin 10, transglutaminase-1, and filaggrin have improved, and the thickness of the epidermis has also grown after two months of in vivo administration of *A. millefolium* extracts compared to placebo [35].

**Figure 2 life-13-00317-f002:**
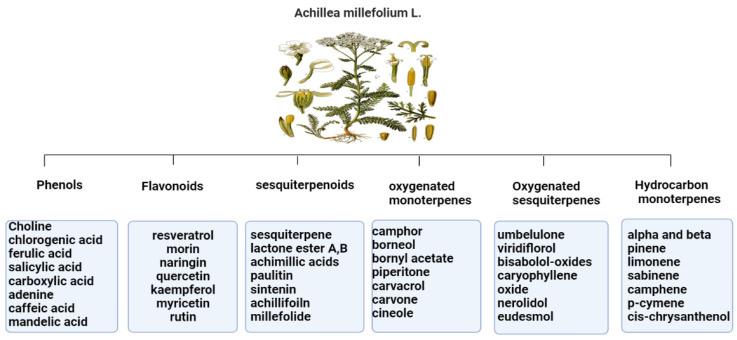
Bioactive phytochemicalcomponents of *Achillea millefolium* L. [36].

### 2.2. Aloe vera

*Aloe vera*, also known as *Aloe barbadensis*, is a member of the *Lilaceae* family. The word aloe comes from the Arabic word “alloeh,” which means “bitter.” *Aloevera* is now used more frequently in the production of new food products due to its medicinal and functional qualities [37]. Early Egyptians referred to *Aloe vera* as the “plant of immortality”, and it has been used as a traditional medicine for more than 2000 years in Arab, Chinese, Egyptian, Greek, Indian, Japanese, and Korean societies to treat diseases including skin issues, constipation, external and internal ulcers, hyperlipidemia, and diabetes [38].

*Aloe vera* production has become a growing industry because of the plant’s many claimed health benefits. It is used to make cosmetic products, laxative medications, and functional foods such as face and hand creams, foundations, cleansers, lipsticks, lotions, shampoos and hair tonics, shaving preparations, bath products, and preparations for makeup and fragrances [39].

Aloe plants have been linked to a variety of biological activities, such as detoxification, treating constipation, flushing out toxins and wastes from the body, and promoting digestion. Aloe plant bioactivities are due to its antibacterial and antimicrobial, antitumor, anti-inflammatory, anti-rheumatoid, and anti-arthritis activities [40]. *Aloe vera* consists of three layers, the outer leaf, green pulp, and aloe gel [41]. Aloe gel, flowers, and leaf skin have all demonstrated antioxidant activities [42]. Since ancient times, *Aloe vera* has been used to cure wounds [43]. *Aloe vera* has been used to treat chronic wounds such as pressure ulcers, as well as burn wounds, surgical wounds, cracked nipples, genital herpes, and psoriasis [44]. *Aloe vera* hydrogel has had a beneficial effect on swelling, angiogenesis, and wound contraction, resulting in a 29% reduction in overall healing time and complete wound closure in just 15 days [45]. Figure 3 shows the bioactive effects of *Aloe vera* on clinical trials obtained during specific time frames.

### 2.3. Curcuma longa

*Curcumin*, a chemical substance that is bright yellow, comes from *Curcuma longa* L. (turmeric) plants of the *Zingiberaceae* family. Approximately 200 years ago, Harvard College laboratory scientists Vogel and Pelletier first identified *Curcumin* in the *Curcuma longa* rhizomes (turmeric) [47]. Traditional herbal medicine has utilized turmeric as a treatment for digestive problems, weight loss, and gastrointestinal and skin inflammation [48]. Turmeric contains curcumin, demethoxycurcumin, and bisdemethoxycurcumin, which are bioactive curcuminoids that have been shown to have anti-inflammatory, anti-cancer, and anti-aging properties [49].

A previous study showed that, following topical *Curcumin* treatment, the wounds of mice closed rapidly with well-formed granulation tissue that was predominantly composed of deposited collagen and a regenerating epithelium. Furthermore, *Curcumin* treatment significantly reduced matrix metallopeptidase-9 and tumor necrosis factor alpha and sped up the healing of wounds in mice via controlling the amounts of different cytokines [50]. Several potential effects of *Curcumin* essential oils are illustrated in Figure 4.

In hairless rats with skin injured by corticosteroids, a combination of *Curcumin* and ginger extract enhances wound healing and skin function simultaneously, lowering the development of non-healing wounds [52].

### 2.4. Althaea officinalis

*Althaea officinalis* L. (Malvaceae), often known as marshmallow, has a long history of use as a medicine to treat laryngopharyngeal mucosal irritation and its associated dry cough. The medicinal plant marshmallow (*Althaea officinalis*) has roots, leaves, and flowers that are frequently used in traditional medicine throughout the world [53]. *A. officinalis* has a variety of substances which have been extracted, including starch, pectins, saccharose, mucilage, flavonoids, caffeic acid, p-coumaric acid, isoquercitrin, coumarins, phytosterols, tannins, and several amino acids [54]. In addition, *Althaea officinalis* is a medicinal herb used to treat lipemia, nose and oral cavity inflammation, and other conditions such as stomach ulcers and platelet aggregation. It has been proven that *A. officinalis* extract displays significant antioxidant activity [55].

For many years, an extract from *Althaea officinalis* has been used to cure wounds and inflammations. The root’s capacity to hold water and its abundance of polysaccharides can boost the immune system [56]. Thetopical application of an *A. officinalis* extract on a rat excision wound model was studied, and the wound healing percentage was much higher in the extract-treated wounds compared to the control, as shown in Figure 5. Moreover, the *A. officinalis* hydroethanolic extract includes phytochemicals that can serve as antibiotics to kill gram-positive bacteria and can quicken the healing of wounds through other mechanisms [57].

### 2.5. Calendula officinalis

Flower extracts from *Calendula officinalis* (pot marigold) have a long history in ethnopharmacology. Traditional medicines for treating mild skin inflammation and promoting the healing of minor wounds include lipophilic and aqueous *Calendula* alcoholic extracts [58]. *C. officinalis* extracts have been linked to several pharmacological actions, the most significant of which include anti-inflammatory, anti-edematous, antioxidant, antibacterial, antifungal, and immunostimulant properties. Terpenoids, flavonoids, phenolic acids, carotenoids, coumarins, quinones, volatile oils, amino acids, and lipids make up the majority of the chemical composition of *C. officinalis* [59]. Other pharmacological properties of *C. officinalis* include antimicrobial, antiviral, effective treatment for breast cancer, antioxidant and anti-immunomodulatory activity, treatment of acne, potent anti-gastric ulcer activity, wound healing properties, treatment of bacterial infections in animals, and hepatoprotective and renoprotective activity [60].

*Calendula officinalis* was used topically and orally to examine its effects on rat excision wounds. The results showed that on the eighth day after the wound was formed, the extract-treated group had a 90.0% wound closure rate (in contrast to the control group’s 51.1% wound closure), and the hydroxyproline and hexosamine contents were significantly higher in the extract-treated group than in the untreated group (Figure 6) [61]. Moreover, *Calendula* ointment can be used to accelerate cesarean recovery since it significantly speeds up the healing of cesarean wounds [62].

### 2.6. Matricaria chamomilla

The Asteraceae family of plants includes the well-known chamomile (*Matricaria chamomilla* L.). The therapeutic and fragrant qualities of German chamomile (*M. chamomilla*) make it a well-known star herb. German chamomile (*M. chamomilla*) is a perennial herb that grows in south-eastern Europe and neighboring Asian nations. Both the flower heads and the essential oils are used in traditional medicine [63]. A wide range of secondary metabolites and types of active compounds are present in *M. chamomilla*, including Sesquiterpenes, polyacetylenes, coumarins, and flavonoids, which are the main ingredients of chamomile. Additionally, luteolin and luteolin-7-Oglucoside, quercetin and rutin, apigenin and apigenin-7-O-glucoside, apigenin and apigenin-7-O-glucoside, and naringenin are among the bioactive phenolic compounds contained in chamomile extracts (Table 2) [64].

The primary uses of chamomile are as an antibacterial, anti-inflammatory, antiseptic, and antispasmodic [65]. Research showed that, within 10 days, re-epithelialization and the development of collagen fibers were accelerated by the use of chamomile extract ointment [66]. Furthermore, chamomile loaded mat is to be suitable for use in wound healing due to its antibacterial, antioxidant, biocompatibility, and mechanical qualities. Moreover, excellent antibacterial effectiveness was demonstrated by 15, 20, and 30% of chamomile loaded mats, and inhibitory zones grew as chamomile content rose. Additionally, these nanofibers exceeded the commercial silver-coated wound dressing in terms of antibacterial efficacy [67].

### 2.7. Eucalyptus

Eucalyptus plantations provide high-quality woody biomass for a variety of industrial applications while alleviating demand on tropical forests and related biodiversity [68]. Due to its quick growth and excellent adaptability to varied settings, Eucalyptus is regarded as a highly successful reforestation tree species [69]. There are plenty of species of Eucalyptus plants [70]. The essential oil extracted from *Eucalyptus globulus* leaves has been used worldwide both as an antiseptic and for easing the symptoms of colds, coughs, sore throats, and other illnesses [71].

In comparison to pure Eucalyptus essential oil (EEO) and regular gentamycin, the optimized nanoemulsion of EEO was chosen for wound healing investigation, collagen estimation, and histological evaluation in rats. The optimized EEO nanoemulsion showed considerable wound healing activity in rats [72]. Other research has shown that *Eucalyptus alba* leaves should be dried at a temperature of no higher than 30 °C and extracted in ethanol for optimal wound healing results and more cell proliferation [73].

### 2.8. Jojoba

The Jojoba plant, which is adapted to a warm, dry climate, is now commercially grown in areas with a severe lack of water, as well as in places where conventional farming methods were previously not financially sustainable. This shrub can withstand high temperatures, requires very little soil fertility and watering, and is also believed to be drought-tolerant [74]. *Simmondsia chinensis*, also known as Jojoba, is a dioecious desert shrub. *S. chinensis* produces seeds that produce liquid wax esters, a valuable ingredient in lubricants used in industry and cosmetics [75].

Jojoba oil is a naturally occurring light yellow oil that can bead ministered topically and is capable of healing wounds and rebuilding skin barriers. Jojoba oil is made up of bioactive substances such as polyphenols, flavonoids, and alkaloids in addition to 97% linear long-chain esters. The administration of Jojoba oil dry nanoemulsion powders (JND) resulted in reduced bleeding and inflammatory cell infiltrations in acute lung injury models (ALI), emphasizing its efficiency as natural oil-based anti-inflammatory and free radical scavengers used to treat ALI [76]. Furthermore, a previous study showed that both keratinocytes’ and fibroblasts’ wound closure are noticeably accelerated by jojoba liquid wax (JLW). JLW was also discovered to promote collagen I production in fibroblasts (Figure 7). JLW’s pharmacological characterization suggests that it could be employed in clinical settings to treat wounds due to its effects on skin cells [77]. Jojoba oil is more commonly referred to as liquid wax than oil since it contains over 98% pure waxes (mostly wax esters, with a small amount of free fatty acids, alcohols, and hydrocarbons), sterols, and vitamins that possess several bioactive properties [78].

### 2.9. Plantago major

The Plantaginaceae family, which includes *Plantago major* L., are also often known as plantain—a perennial herb with rosette-shaped leaves that are 15 to 30 cm in diameter [79]. The plantain plant, or *Plantago major*, thrives in a variety of habitats, including roadside ditches, developed fields, canal water, and waste sites [80]. The plant leaves and seeds have a long history of usage in folk medicine for a variety of ailments, including the treatment of a wide range of illnesses and disorders including digestive problems and respiratory issues. Additionally, it has been utilized as an anti-inflammatory, anti-microbial, and anti-tumor agent to treat wounds. Plantains also contain compounds that can counteract both internal and external toxins [81].

*Plantago major* contains a number of active substances, including caffeic acid derivatives, polysaccharides, terpenoids, lipids, and flavonoids, as is shown in Table 3 [82]. A safe and effective herbal remedy for the treatment of second-degree burn wounds is *P. major* ointment, which not only contains qualities for healing wounds but also functions as an analgesic and an antibacterial agent [83]. Furthermore, it was found that, when compared to alternative therapies, *P. major* extract effectively accelerated the healing of dorsal cervical injuries in mice [84]. *P. major* leaves can accelerate wound-healing processes in an ex vivo pig wound-healing model. That extracts of freeze-dried leaves made from both ethanol and water have shown a stimulating effect suggests that *P. major* might be a useful source of many bioactive compounds with wound healing potential. Moreover, a concentration of 1.0 mg/mL (dry weight) has the best results for both types of extracts [85].

### 2.10. Inula

*Inula* is a vast genus of about 100 species of flowering plants that are native to Europe, Asia, and Africa. Many of these species have been used as sources of medicines [86]. *Inula* contains several terpenoids, flavonoids, and lignins. Sesquiterpenoids from the genus *Inula*, including eudesmanes, xanthanes, dimers and trimers of sesquiterpenoids, are also widely distributed [87]. Many different biological effects have been observed for chemicals derived from *Inula* species against oxidative stress-related illnesses, inflammation, diabetes, cancer, and neurological diseases [88].

A perennial herbaceous plant called *Inula viscosa*, often locally known as “Magramane”, is found throughout the Mediterranean basin. As a result, the plant has been employed in traditional medicine to treat a variety of illnesses; this is attributed to its anti-inflammatory, anthelmintic, antipyretic, antiseptic, and antiphlogistic properties [89]. *Inula viscosa* leaves could serve as a source of bioactive substances, volatile oils and phenolic compounds. Additionally, its potential for the development of natural preservatives with applications in agro-food is indicated by its antioxidant, antibacterial, and antifungal capabilities [90]. The *Inula* genus is a source of structurally diversified antioxidant chemicals that can combat many oxidative stress-related human diseases via various pathways, making it beneficial for the creation of new medications [91]. The administration of extracts of *Inula racemose* (Ir) significantly increases total phenolic content and free radical scavenging activity, and the protein expression of p53, bax, and bcl-2 are returned to near-normal levels. Thus, the extracts of Ir may be potential therapeutic agents for providing several beneficial effects in hepatic injury following orthotopic liver transplantation [92].

The pharmacological wound healing bioactivities possessed by *Althaea officinalis*, *Calendula officinalis*, *Matricaria chamomilla*, Eucalyptus, *Plantago major*, Jojoba, and *Inula* are presented in Figure 8.

### 2.11. Pine

To produce a variety of products, pine trees are planted all over the world. For building materials, there are wood and cellulose [93]. Pine produces phenolic chemicals and oleoresin, the latter of which contains a variety of terpenoids. Polyphenolic parenchyma cells and specialized ray cells both create phenolic chemicals, and specialized resin ducts are distributed throughout the wood, bark, roots, and needles. Specialized secretory tissues also create large amounts of terpenoid resin, which are then stored there [94].

The pine tree is constituted of several parts, including wood, bark, resin, needles, cones, and seeds [95]; the chemical composition of each part of the pine tree are shown in Figure 9.

Pine pollen polysaccharides (PPPS) can speed up the recovery of mouse skin wounds and encourage the growth of chicken embryo chorioallantoic vasculature by encouraging cell division, shifting the cell cycle from G1 to S and G2, and upregulating Cyclin B1 expression in vitro. These effects of PPPS were accomplished by JAK2-STAT3 signaling pathway activation [96]. The largest benefits in the wound healing activity models were seen with the essential oils extracted from *Pinus pinea* and *P. halepensis* cones. However, other essential oils did not exhibit any notable anti-inflammatory or wound healing properties [97]. Abietic acid makes up more than 50% of pine resin. Abietic acid significantly increased angiogenic capacity, which is linked to increased production of p38 and extracellular signal-regulated kinase (ERK). Furthermore, abietic acid accelerated cell migration and tube formation in human umbilical vascular endothelial cells. In mouse models with cutaneous wounds, groups treated with 0.8 μM abietic acid demonstrated accelerated wound closure in comparison to the control groups [98].

**Figure 9 life-13-00317-f009:**
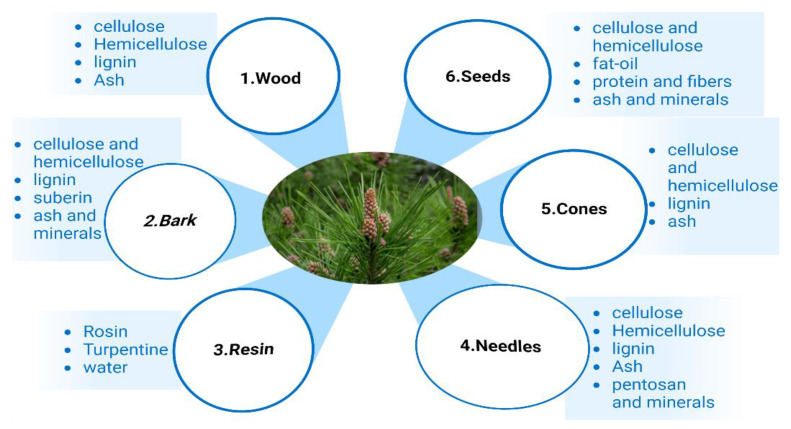
Chemical composition of different pine parts [99].

### 2.12. Green Tea

More than two-thirds of the world’s population drink tea, including green tea brewed from the leaves of the *Camellia sinensis* plant [100]. One of the oldest and most widely consumed beverages worldwide, it is made from the *Camellia sinensis* (L.) plant (green tea) and is mostly grown in Japan, China, and Taiwan [101]. Green tea has been demonstrated to have positive impacts on human health, including effects against cancer, obesity, diabetes, heart disease, infections, and neurological disorders [102].

Numerous variables, including agricultural methods, climate, season, and plant types, can affect the content of tea. Green tea’s key ingredients are polyphenols, mainly flavonoids. Catechins make up 6%–16% of the dried green tea leaves. The four main catechins are: epigallocatechin (EGCG), which accounts for roughly 59% of the total number of catechins; epigallocatechin (EGC), accounting for roughly 19%; epicatechin-3-gallate (ECG), accounting for roughly 13.6%; and epicatechin (EC), accounting for roughly 6.4%amount [103].

Numerous studies have been conducted on the health advantages of green tea, especially epigallocatechin gallate(EGCG); it iswell recognized that these effects are mostly related to its polyphenols [104]. The most prevalent component in tea leaves is EGCG, which is thought to have the major bioactivities (Figure 10). These bioactivities include free radical scavenging properties, antimicrobial, anti-inflammatory, and angiogenic effects that induce a proper wound healing process and minimize the onset of infection [105].

Episiotomy pain seems to be effectively reduced by green tea ointment, which also helps to speed up wound healing [107]. Moreover, animal experimentation and molecular mechanism studies have demonstrated that green tea polyphenols could speed up diabetic rats’ ability to heal wounds by modulating the PI3K/AKT signaling pathway [108].

### 2.13. Punicagranatum L.

Pomegranate, known as *Punica granatum* L., is an old fruit full of bioactive components including total phenols, flavonoids, hydrolysable tannins, proteins, vitamins, and minerals [109]. A pomegranate fruit consists of several parts: the peel, which accounts for 49% to 55% of the total fruit size, and the arils, which account for45% to 52% (18% to 20% seeds and 26% to 30% juice) [110]. Pomegranate seeds are a rich source of phytochemicals with a high antioxidant activity [111]. It has been shown that pomegranate peel extracts had significant positive effects in the minipig second-degree burn model, which may be related to increases in VEGF-A and TGF-b1 protein and gene expression levels [112]. Pomegranate is a promising medicinal plant for the recovery of skin burn wounds, where previous research showed that 10% of standard pomegranate extracts can hasten the healing of serious second-degree burn wounds which is characterized by angiogenesis, a complete and mature epithelium, a low number of inflammatory cells, and a high density of collagen with a good organization [113].

Moreover, rats given cream containing *P. granatum* flower extract experienced faster wound healing on day 25 of the treatment than those given other treatments—where the wound healing was accelerated by the *P. granatum* flower extract, which can also be utilized to treat burn injuries [114]. According to mechanical (contraction rate, tensile strength), biochemical (raising of collagen, DNA, and protein synthesis) and extract assessments, pomegranate peel ointment considerably improved the wound contraction and the time of epithelialization in excised wounded models over 10 days [115]. The pharmacological bioactivities of various medicinal plant species are summarized in Table 4.

## 3. Conclusions

Each of the botanicals we have studied possesses chemical complexities that are connected to several pharmacological activities. These specifically shared characteristics include anti-microbial, anti-inflammatory, and anti-oxidative properties. Future medications may emerge from medicinal herbs, which have fewer side effects and better bioavailability for healing wounds. Particularly when treating patients with persistent or resistant skin ulcers, topical or systemic usage of phytotherapeutic drugs alone or as a supplement to other healing medications should be considered.

## Figures and Tables

**Figure 1 life-13-00317-f001:**
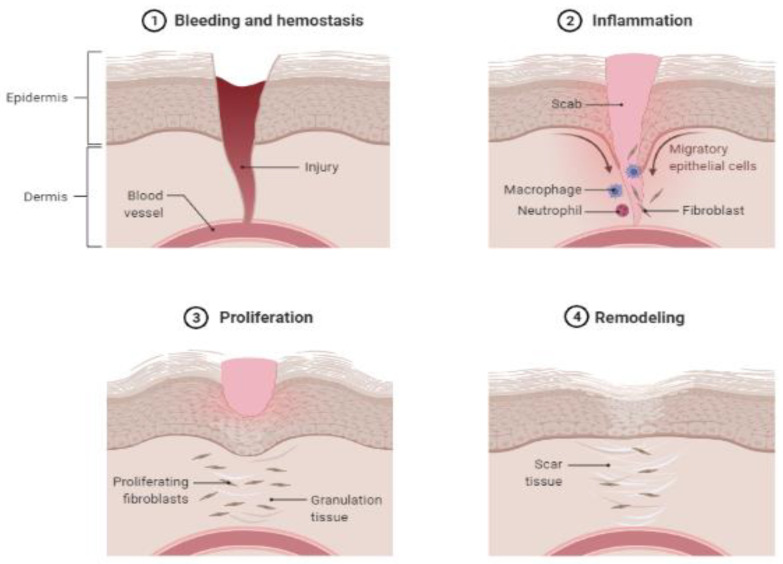
The four main phases of the wound healing process [6]. Adapted from https://app.biorender.com/biorender-templates2023/figures/all/t-5fa1b1622a60ac00a3d858ec-wound-healing (accessed on 15 December 2022).

**Figure 3 life-13-00317-f003:**
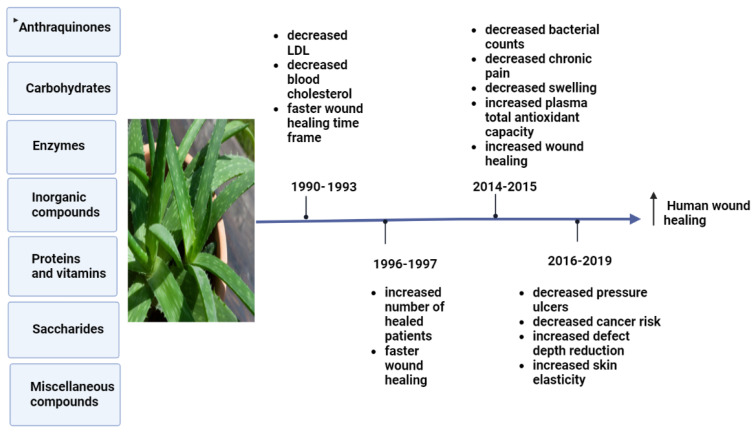
Effects of pharmacological bioactive components of *Aloe vera* on clinical trials [46].

**Figure 4 life-13-00317-f004:**
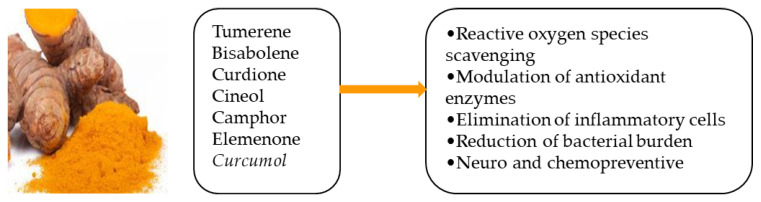
Some active components of curcumin essential oils and their potential effects [51].

**Figure 5 life-13-00317-f005:**
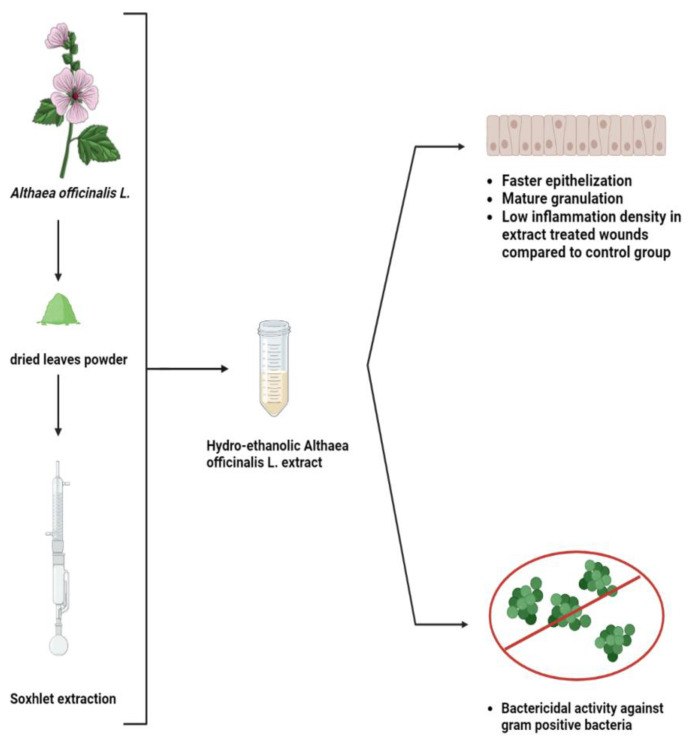
Efficiency of *Althaea officinalis* L. extract that significantly heals excision wounds on rats and inhibits gram positive bacteria compared to control group [57].

**Figure 6 life-13-00317-f006:**
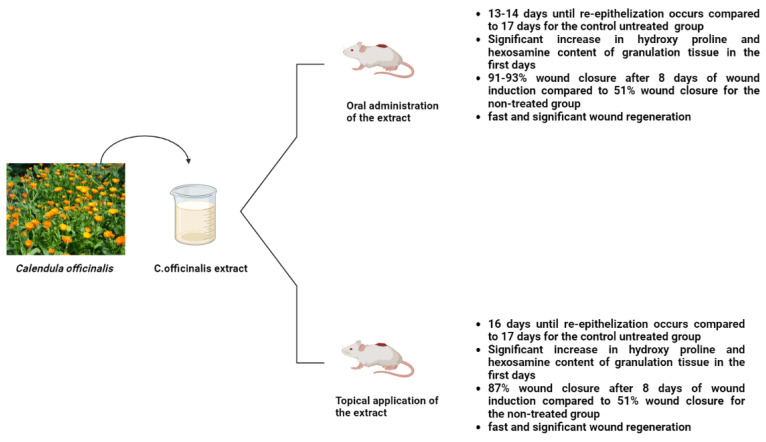
Effects of *Calendula officinalis* extract on wound closure, regeneration, hydroxyl proline, and hexosamine content in rat models [61].

**Figure 7 life-13-00317-f007:**
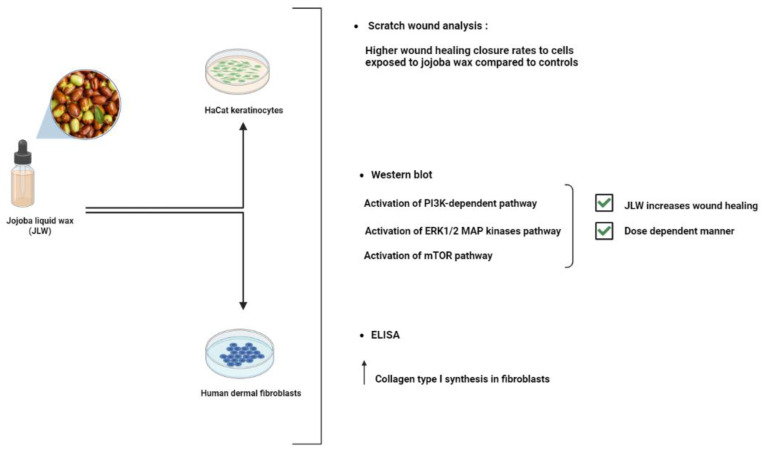
Jojoba liquid wax (JLW) wound healing properties examined by western blot, ELISA and scratch wound analysis on in vitro human dermal fibroblasts and keratinocytes [77].

**Figure 8 life-13-00317-f008:**
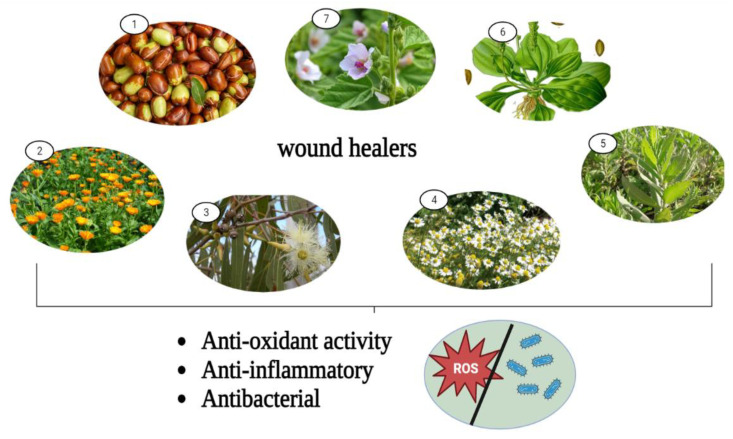
Pharmacological wound healing activities of some remarkable medicinal plants. 1: Jojoba, 2: *Calendula officinalis*, 3: Eucalyptus, 4: Chamomile, 5: *Inula*, 6: *Plantago major*, 7: *Althea officinalis*.

**Figure 10 life-13-00317-f010:**
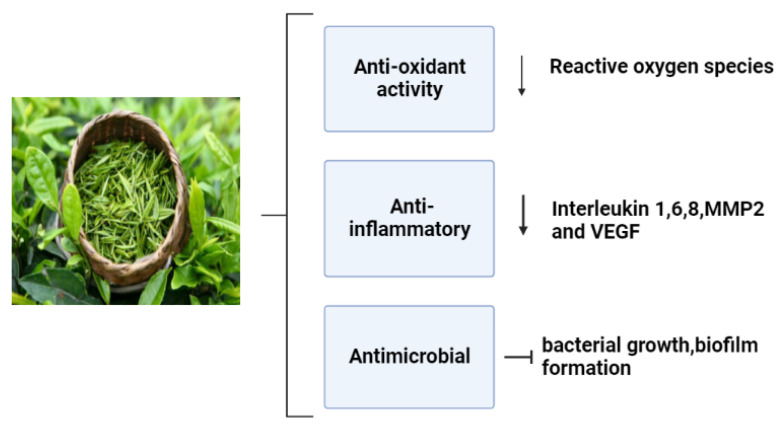
Pharmacological effects of green tea on the wound healing mechanism [106].

**Table 1 life-13-00317-t001:** Dermatological problems treated by various traditional herbal medicines.

Medical System	Dermatological Conditions	Reference
Chinese	Diabetic foot ulcers	[19]
Ayurveda	Psoriaisis	[20]
Unani	Pityriasis versicolor	[21]
Russian	Vitiligo, psoriasis	[22]

**Table 2 life-13-00317-t002:** The phytochemistry of the major bioactive components of *Matricaria chamomilla* L. [64].

Molecule Name	Chemical Structure
Chlorogenic acid	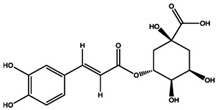
Caffeic acid	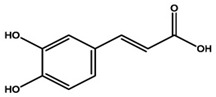
Luteolin	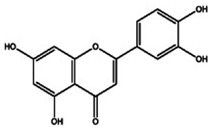
Apigenin	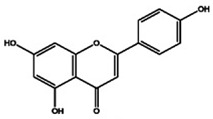
Naringenin	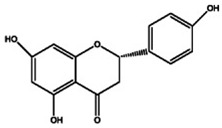
Rutin	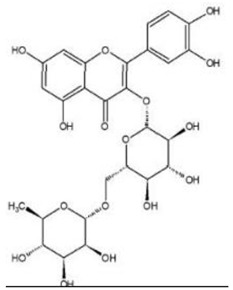
Quercetin	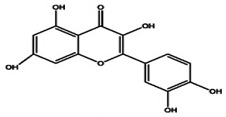

**Table 3 life-13-00317-t003:** The phytochemistry of the major bioactive components of *Plantago major* [82].

Molecule Name	Chemical Structure
Phenylethanoid glycosides	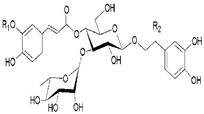
Triterpenoids	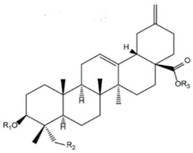
Caffeic acids	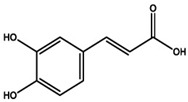
Coumarins	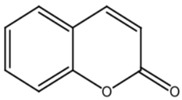
Polysaccharides	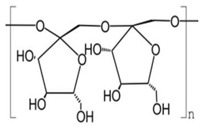
Phenolic acids	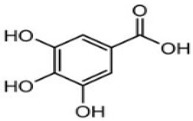
Sterols	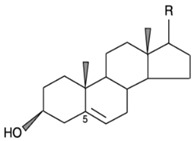

**Table 4 life-13-00317-t004:** Pharmacological effects of some medicinal plants.

Plant Species	Pharmacological Activities	Reference
*Achillea millefolium* L.	Antimalarial, antioxidant, antiulcer, Antispasmodic, antihypertensive, hepatoprotective, gastroprotective, antimicrobial, anticancer, anti-inflammatory, analgesic effect, skin rejuvenating activity	[36]
*Aloe vera*	Anticancer, antimicrobial, cardioprotective effect, antidiabetic, digestive diseases protection, skin protection, prebiotic activity, bone protection, anti-inflammatory	[116]
*Althaea officinalis*	Anti-inflammatory, anti-cough, anti-bacterial anti-fungal, immunostimulatory, antioxidant, wound healing	[53]
*Calendula officinalis*	Anti-inflammatory, antioxidant, spasmogenic effects, neuropharmacological remedy	[117]
*Matricaria chamomilla*	Anti-inflammatory, anti-microbial, antiparasitic, antioxidant, anticancer, analgesic, anti-diabetic, anti-anxiety, gastroprotective effect, antibacterial	[118]
*Curcumin*	Antioxidant, anti-inflammatory, bactericidal wound contraction, anticancer, antiasthma, skin health promotion, jaundice, antidiabetic	[119]
*Eucalyptus*	Antioxidant, antibacterial, neuroprotective anti-ischemic, anti-hypertensive, antiviral	[120,121,122,123]
*Jojoba*	Antioxidant, antiviral, antimicrobial, hepatoprotective, antiglycemial, analgesic anti-inflammatory, transdermal drug delivery	[124]
*Plantago major*	Wound healing, antidiabetic, anti-inflammatory, anti-bacterial, antiviral, antioxidant, anti-ulcer	[125]
*Pine*	Antimicrobial, antioxidant, cardiovascular, neuroprotective, anti-inflammatory, anticancer	[126]
*Green tea*	Antioxidant, anti-inflammatory, antimicrobial, angiogenesis stimulation, immunomodulation, anticancer, antidiabetic, hypoglycemic	[127]
*Punicagranatum*	Antioxidant, anti-inflammatory, anti- bacterial, antimicrobial, anti-cancer, wound healing, gastrointestinal diseases protection	[128]
*Inula*	Antioxidant, antihyperglycemic, antimicrobial, anti-cancer, anti-inflammatory	[129]

## Data Availability

Not applicable.

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
