# Peer review of "The Therapeutic Wound Healing Bioactivities of Various Medicinal Plants"

_life, 2023, doi:10.3390/life13020317_

Round 1

Reviewer 1 Report

Title: The therapeutic bioactive wound healing potentials of various medicinal plants and herbs.

Revise....potentials? and difference between medicinal plants and herbs?

Abstract

These sentences could be combined: The skin serves as the body's main line of defense, guarding against mechanical, chemical, and thermal damage to interior organs. This defense also includes a highly developed immune response that serves as a barrier against pathogenic infection.

When the skin is damaged, germs can quickly enter the tissues beneath the skin, which can result in infections that are potentially fatal. Revise 'infections that are potentially fatal'

Revise statement: Therefore, to cope with such 'pathological situations', appropriate remedies are required. Botanicals have been used effectively to treat chronic and infected wounds where the treatments offered by phytomedicine have considerable pharmacological effects that have widely been employed for 'wounds treatment'

Identify the geographical regions where these plants are endemic: Achiella millefolium L., Aloe vera, Althaea officinalis, Calendula officinalis, Matricaria chamomilla L., Curcuma longa, eucalyptus, jojoba, plantain, pine, green tea, Punica granatum, inula.

Revise: The use of phytotherapy may create new opportunities for...

Again, attempt to identify the region/s where such knowledge and use occur: This review provides the most often used medicinal plants that are helpful in the treatment of wounds and suggests viable alternatives for the wound care field.

The whole manuscript has several syntax issues that can only be solved using professional English-editing services. I urge authors to approach a credible English-editing service provider before their manuscript can be considered for publication in this or any other journal. I have provided examples of sentences that require attention - but only in the title and abstract.

Author Response

Dear reviewer,

Thank you  for your comments.

Please find the answers to remarks in attached file.

Best regards,

Authors,

Reviewer 2 Report

- Try to shorten the long paragraphs you have in the introduction section

- can you reveal the chemical structure of some important bioactive chemicals and compare with the existing modern medicine

- try to compare the wound healing potential of the plants or their active components 

Author Response

Dear Reviewer,

Many thanks for your judicious comments that allow us to improve our manuscript.

Please find the answers to comments in attached file.

Best regards,

Authors,

Reviewer 3 Report

Paper is well written, however paper is not well suited for this journal as it is a review and these plants have been extensively researched. also i dont seem to find any novelty of this review paper as all the said plants have already been published in reputed journals. 

you can add more figures and table for its innovativeness which included different phytochemistry for wound healing, or the mechanismd of these plants for wound healing. 

paper seems to be average for the publication in thus journal.

Author Response

Dear reviewer,

Thank you so much for you judicious comments which allow us to ameliorate our manuscript.

Please find answers to comments in attached file.

Best regards,

Authors

Round 2

Reviewer 1 Report

Comments

There have been improvements in the structure and language of the manuscript. However, I could still not see evidence that the manuscript was edited by a first speaker of English as advised. Or, at best, the authors rushed the revision through and ignored the apparent errors listed below. What is listed below is not an exhaustive list of improvements to be made. I recommend further editing – not limited to what is listed below. The correctness of the manuscript is the responsibility of the authors. And it is not the duty of the reviewers to edit the manuscript on behalf of the authors. I, therefore, recommend that the authors take their time and look at each sentence, paragraph, section, etc. and resubmit for publication.

Abstract:

1.      The skin serves as the body's main line of defense, guarding against mechanical, chemical, and thermal damage to the interior organs, which includes a highly developed immune response that serves as a barrier against pathogenic infection….this sentence needs splitting.

2.      Use better scientific term for germs

3.       Natural phytomedicines, that possesse considerable pharmacological properties, have been…spelling of ‘possesse’

4.      Natural phytomedicines, that possesse considerable pharmacological properties, have been widely and effectively employed on wound treatments and infection prevention Achiella millefolium Aloe vera, Althaea officinalis, Calendula officinalis, Matricaria chamomilla., Curcuma longa, Eucalyptus, Jojoba, plantain, pine, green tea, pomegranate and Inula are the most remarkable wound 32 healing botanicals used in the Northern hemisphere….this long statement must be revised.

Main

5.      They are two main types of wounds which are acute wounds that go through 53 an orderly, prompt healing process. These cancan lead to a long-lasting 54 restoration of anatomical and functional integrity….revise these

6.      outcome, or have failed to do so in a 57 timely and arranged manner…revise

7.      Figure 1. The four main phases of an appropriate wound healing process [6]….revise, delete appropriate

8.      Platelets have -granules in their cytoplasm … revise

9.      After that, Keratinocytes, fibroblasts…

10.   The extra cellular matrix basement…

11.   Line 91 …. The extra cellular matrix basement…revise ECM? And then use ECM only subsequently…

12.   The introduction is rather too long (At least 60 lines!), there is nothing new about all the information provided here. Cut. Authors need to quickly migrate their argument to the subject matter – introduce the wound healing properties of plants.

13.   Complementary and alternative medicine is used to treat non-serious minor…

14.   wellness, and fend off diseases, this is due in part 117 to medicinal plants that involve plenty of bioactive ingredients

15.   This part could come in earlier in the introduction…revise. By 2024, the market for wound care products is anticipated to grow economically to around $15–22 billion annually. Wounds are now a category in the National Institutes of Health's Research Portfolio Online Reportin Tool; due to the rising health care expenses, an aging population, awareness of infection hazards including biofilms that are challenging to treat, and the ongoing danger of diabetes and obesity worldwide [25].

16.   Line 148

17.   Line 156….concludes the introduction? A good introduction for the current paper must not exceed 500 to 600 words. Authors exceeded 1000 words.

18.   Line 180… Flavonoids like kaempferol, luteolin, and apigenin are among of 180 Achillea's primary ingredients

19.   Line 182… revise figure 2 to Figure 2.

20.   Specify…. The majority of this herb's antioxidant

21.   Line 214..

22.   Line 230- Authors discuss the functions of curcumin, which is not a plant. Instead the section should be titled Curcuma longa… revise.

23.   Repetition - For many years, people have used an extract from the marshmallow 266 plant, Althaea officinalis, to cure

24.   Line 272, correct figure 5 (do the same elsewhere).

Author Response

Dear Reviewer,

Thank you so much for your valuable comments.

Please find the answers to comments in the attached file.

Best regards,

Authors

Reviewer 3 Report

all comments were taken into consideration.

Author Response

Dear Reviewer,

Answer : Thank you very much for your encouraging remarks.

Best regards,

Authors